# Spatial Distribution of Tumor Cells in Clear Cell Renal Cell Carcinoma Is Associated with Metastasis and a Matrisome Gene Expression Signature

**DOI:** 10.3390/cancers17020249

**Published:** 2025-01-14

**Authors:** Prahlad Bhat, Pheroze Tamboli, Kanishka Sircar, Kasthuri Kannan

**Affiliations:** 1Department of Translational Molecular Pathology, The University of Texas MD Anderson Cancer Center, 2130 W Holcombe Blvd., Houston, TX 77030, USA; prahladbhat@utexas.edu; 2Department of Pathology, The University of Texas MD Anderson Cancer Center, 1515 Holcombe Boulevard, Houston, TX 77030, USA; pheroze.tamboli@mdanderson.org

**Keywords:** clear cell renal cell carcinoma, metastasis, matrisome, spatial, point process, digital pathology, tumor grade

## Abstract

Clear cell renal cell carcinoma (ccRCC) is the most common type of kidney cancer, but predicting its behavior remains challenging using standard histopathologic examination. This study introduces a novel approach to predict ccRCC aggressiveness by analyzing the spatial distribution of tumor cells in H&E-stained images. The researchers found that spatial analysis outperformed traditional tumor grading in predicting metastasis, particularly for intermediate-grade tumors. They identified two distinct patient groups based on spatial characteristics, with one group showing greater spatial randomness and a higher association with metastasis. Furthermore, the study revealed a gene expression signature related to the extracellular matrix (matrisome) that correlated with the spatial patterns and aggressive tumor behavior. These findings suggest that analyzing the spatial distribution of ccRCC tumor cells could provide valuable insights into tumor behavior and metastatic potential, potentially improving prognostication and personalized treatment strategies for patients with ccRCC.

## 1. Introduction

Among renal cell carcinomas (RCCs), clear cell RCCs (ccRCCs) represent the most common subtype, accounting for 70% to 80% of all RCCs [1]. Owing to the increased use of imaging studies, the detection and incidence of RCC have risen steadily over the past few decades. However, there has not been a corresponding rise in RCC mortality, as many incidentally detected RCCs are indolent [2]. Differentiating aggressive RCC from indolent RCC remains a major challenge in patient management, and no prognostic biomarkers are currently in use beyond standard clinicopathologic parameters such as tumor grade and stage. Clear cell RCC tumors are graded on a 4-tier scale (grades 1–4) according to the International Society of Urological Pathology/World Health Organization (ISUP/WHO) system. Tumor grading in RCC, which involves histological assessment of a tumor cells’ nuclear characteristics, is independently prognostic, particularly for grade 4 ccRCC [3,4]. Furthermore, sarcomatoid differentiation in RCC—a subset of grade 4 RCC—wherein tumor cells lose their epithelioid morphology and acquire a mesenchymal sarcoma-like morphology, is prognostic and detectable by histology.

RCC tumor grading, however, has significant drawbacks that limit its utility as a prognostic tool in most RCC cases. First, RCC tumor grade is assigned based on the highest-grade component within a tumor, which may span only a few high-power microscopic fields. Preoperative renal mass biopsy tends to yield lower grades compared to the resected specimen [5], given the heterogeneity of different RCC grades and the limitations of sampling only a small portion of the tumor that may not include the highest-grade components, including sarcomatoid features. For this reason, grade 4 ccRCC tumors, while prognostic, may not be detected following a biopsy. Second, while true grade 1 ccRCCs show excellent prognosis, they are exceedingly rare. Indeed, the large majority of ccRCC tumors are of grades 2 or 3 [6], and standard histological grading performs poorly in this cohort. Studies have shown that patients with grade 2 and grade 3 ccRCC to have overlapping survival curves [7]. Accordingly, predicting the metastatic potential of grade 2 and grade 3 ccRCC is clinically relevant but also challenging using only nuclear morphology.

Given that infiltrative processes mediated by coordinated rearrangements of the cells’ distributions are vital for tumor progression, we sought to incorporate spatial characteristics of the tumor as a predictor of disease aggressiveness. Specifically, we explored the utility of spatial point pattern measurements from H&E-stained images of grade 2 and grade 3 ccRCCs. We found that spatial modeling of tumor cells distinguishes the metastatic phenotype, outperforms International Society of Urological Pathology/World Health Organization (ISUP/WHO) grading, and is associated with extracellular matrix genes that may drive these tumors’ aggressive behavior.

## 2. Materials and Methods

### 2.1. Sample Collection

This retrospective study was performed with informed consent and using an institutional review board-approved protocol (IRB# LAB 08–670). The MD Anderson-derived ccRCC tissues submitted to The Cancer Genome Atlas (TCGA) were processed via standard formalin fixation and paraffin embedding techniques, as illustrated by Sadeghipour et al. [8]. All the derived whole-slide images of the formalin-fixed paraffin-embedded tissues came from the TCGA GDC portal, where we queried the diagnostic slides. Inclusion criteria included ccRCC and ISUP/WHO histologic grades 2 or 3. We obtained H&E-stained images from 72 patients, nearly half of whom had metastatic disease. Table 1 summarizes the patients’ tumor grade, pathologic stage, survival status, and metastatic status. Appendix A lists the specific TCGA identifiers.

### 2.2. Study Workflow

The study workflow, as illustrated in Figure 1, consisted of identifying tumor cells with a purity of at least 75% and generating spatial point patterns. We generated 3 spatial point pattern objects for each patient, constituting 216 regions of interest, and used QuPath (v0.4.3) [9] to annotate the tumor cells and derive tissue boundaries.

### 2.3. Identifying Tumor Cells

The identification, segmentation, and classification of tumor cells were conducted entirely in QuPath [10]. Once diagnostic slides were queried and loaded into QuPath and several layers of annotations were created by a subspecialized genitourinary pathologist. Using the Brush tool, a “master annotation” encompassing the area of the entire tumor containing high-purity (75% tumor purity) components was drawn (Figure 2). The Create Region Annotation tool was then used to generate 3, 750 µm × 750 µm “random annotations” (RAs) that were scattered randomly across the master annotation. Whole-slide images in this study were scanned at 50× (0.25 µm/pixel) or 25× (0.5 µm/pixel), and 50× images were down sampled by a factor of 2 to ensure standard resolutions. Within each RA, necrotic masses, blood vessels, and image artifacts that created undesired white space were cleaved using QuPath’s Pixel Thresholder tool. Once unwanted objects were erased from the RA, the individual cells were segmented using the StarDist segmentation plugin [11]. Because ccRCC cells are highly vascular and cell boundaries are often difficult to discern [12], the nucleus of a cell was assumed to be the cell’s center for the purposes of this study. After cell segmentation on each RA, a new set of cell-level annotations was introduced in order to isolate only tumor cells. Two sets of annotations were made per RA—non-tumor and tumor training annotations—in order to build a unique model that could be used to classify the remaining cells in each RA. Non-tumor cells included endothelial cells, stromal cells, and inflammatory cells. The model was constructed using the Object Classifier and selecting the Random Trees option, which uses staining intensities and morphometric measures of segmented cells to assign cells using training annotations as input. Each RA had its own segmentation model. A pathologist annotated at least 10 tumor and 10 non-tumor cells and ran the Object Classifier, adding more annotations until the classification accuracy did not significantly improve; this reduced over- and under-fitting. This process was repeated for each tissue sample. The annotations and the scripts are provided in a Github repository (see Data availability section).

### 2.4. Generating Spatial Point Pattern Objects

QuPath converted the nucleus segments into cartesian coordinates (detections), which represented the centroid of the elliptical nucleus. The origin (0,0) was located at the top left corner of the image. Once cells were segmented and classified for all tissues, three types of point measurements were exported from QuPath: a matrix of points that represented the vertices of the square RA, a separate matrix of points within the RA that traced the shape of the cleaved segments, and a matrix of points representing the nucleus detections contained within the RA. The first two matrices were merged by creating a square window object and creating holes using the cleaved segment coordinates. Next, the points were merged with the bounded region to create a spatial point process. This spatial point process was split into two subprocesses: one containing only tumor points and one containing only non-tumor points. The tumor process was isolated for downstream analysis.

### 2.5. Spatial Statistics and K-Means Analysis

We obtained the pair-correlation function or PCF statistic for each spatial point pattern. The PCF, g(r), is the probability of observing a pair of points r units apart divided by the probability if the points exhibit complete spatial randomness [13]. It is defined by$$g(r)=\frac{K^{\prime }\left(r\right)}{2\pi }$$
where K′(r) is the derivative of Ripley’s K-function. We computed the PCF along a discrete interval of [0.5 µm, 35 µm] with increments of 0.5 µm consisting of 70 observations. We averaged the PCF values for each patient (PCFs from three spatial point pattern domains) and each r, resulting in a 72 × 70 matrix. This matrix was subject to K-means analysis for K ranging between 1 and 10. We then used the average silhouette width for each K to determine the optimal number of clusters.

### 2.6. Geyer Saturation Process Simulation

The Geyer Saturation Process (GSP) provides a valuable framework for modeling the spatial interactions within point pattern data sets, offering insights into the spatial dependencies (both inhibition and clustering) between cells [14]. First, we added homogeneous Poisson distributed random cells to each spatial pattern based on the intensity of the cells in the point pattern. These simulated points were marked as “random cells”, while the original points were marked “cells”. Next, we generated 500 realizations of each point pattern of each mark (i.e., the mark could be cells or random cells) by subsampling the points based on Cochran’s sample size estimator [15]. We obtained multiple GSP fits for each realization based on several input parameters. The GSP algorithm requires two parameters: radius r and the saturation parameter. We fixed the saturation parameter as either 1, 2, or 3. We computed the vector of nearest neighbor distances to choose the input radii. These parameters resulted in multiple fits for each realization, and we extracted the optimal fitting parameters based on Akaike information criterion [16]. We extensively used the R spatstat library [17] for these simulations, especially the profilepl function, which can generate multiple GSP fits and output the optimal fit based on a grid of radii and saturation parameter values.

### 2.7. Gene Expression Analysis and Multi-Institutional Cohort Validation

We downloaded the RNAseq data from the TCGA-KIRC data set using the R package TCGAbiolinks [18] for all the patients for whom the data were available (71 patients). The unstranded raw gene counts were queried and downloaded for all patient IDs, detailed in Appendix A. We batch-corrected the data in ComBat [19] and used DESeq [20] to perform differential gene expression analysis between the Group 1 and Group 2 patients. Finally, we used the absolute threshold of 3 for the log2 fold-change in differentially expressed genes to query a larger cohort of 352 ccRCC patients from multiple institutions in the cBio portal [21]. We investigated the clinical and survival characteristics of patients with alterations in the resulting differentially expressed genes.

## 3. Results

### 3.1. Spatial Analysis Reveals Two Patient Groups Segregated by Spatial Randomness and Metastatic Status in ccRCC

Silhouette width analysis for various K-means aggregations of patients, obtained by clustering the PCF values, showed that the optimal number of groups was *n* = 2 (Figure 3A). This separation into two groups was in agreement with the principal component analysis (PCA) plot that showed two clear segregation of PCF values when reduced to two dimensions (Figure 3B). Figure 3C shows the PCF as a function of the radii r with the corresponding error bars. The closer PCF values are to 1—over a range of radii—the greater the propensity for spatial randomness. Therefore, as indicated by the error bars in Figure 3C, Group 1 exhibited more spatial randomness compared to Group 2, at r > 8 µm. Notably, spatial Group 1 also showed significantly greater association with clinical metastasis compared with spatial Group 2 (*p* < 0.01), whereas histologic ISUP/WHO grade showed no association with metastatic status (*p* = 0.79, Figure 3D). Appendix A lists the patient identification numbers and the spatial group to which they belong.

### 3.2. Aggressive ccRCC Samples Show Greater Inter-Cellular Distances Between Tumor Cells

A critical output parameter for the optimal fits in GSP is the interaction radius r. While PCF gives the normalized probability that a certain distance separates two cells, the interaction distance parameter for the GSP gives the actual distance of interaction between the cells, and therefore we can compare this to the PCF, as both are local measures. Consistent with the PCF analysis, the mean interaction distances between tumor cells in Group 1 patients were significantly higher than those in Group 2 patients (Figure 4A). The mean interaction distances between tumor cells in all patients with metastatic disease and in Group 1 patients were greater than the intercellular distance among simulated random cells (Figure 4B,C). Interestingly, the mean interaction distances between tumor cells in Group 2 patients did not differ significantly from those in random cells (Figure 4D). These results suggest a high degree of tension between the cells in the Group 1 patients as well as those with aggressive, metastatic disease, potentially resulting in the loss of cell adhesion and an eventual metastatic cascade.

### 3.3. Gene Expression Differences Between Spatial Groups Reveal a Matrisome Signature

We identified a 399-gene signature with an adjusted *p*-value cut-off of 0.01 for differential gene expression analysis between Group 1 and Group 2 (Appendix A). Investigating these genes for signaling pathways revealed a matrisome (extracellular matrix) signature of 43 genes and a core matrisome signature of 16 genes (Appendix A). Figure 5 shows the volcano plot for the 156 genes with the absolute log2 fold-change greater than 3, including 21 matrisomal genes.

### 3.4. Spatially Defined Gene Expression Signature Stratifies a Multi-Institutional Cohort of ccRCC Patients by Survival and Stage

Differential gene expression analysis on a larger multi-institutional TCGA-KIRC cohort revealed significant survival differences (overall, disease-free, and progression-free) between patients who had alterations in these 156 genes and those who did not. Moreover, there was significant stratification by disease stage for these gene-altered patients. Figure 6A,B show the progression-free survival and the stage segregation. Appendix A shows the overall and disease-free survival curves.

## 4. Discussion

The results presented in this study provide valuable insights into the intratumoral cellular dynamics, shedding light on cell-to-cell interactions and metastasis. Identifying two distinct patient groups through K-means analysis based on PCF values provides a compelling starting point. An over-representation of patients with metastatic disease in Group 1 compared to Group 2 establishes a rationale for further statistical modeling and simulations. The implications of these findings extend beyond statistical correlations, hinting at potential mechanisms underlying metastatic progression in ccRCC patients. The fact that the spatial distribution of tumor cells alone distinguishes a group of patients with metastasis raises intriguing questions about the underlying biological relevance of this spatial distribution. Furthermore, the lack of stratification of patients with intermediate-grade metastatic disease based on standard-of-care pathological grading reinforces its weaknesses as a prognostic parameter.

Herein, we propose a novel method for predicting the behavior of ccRCC tumors from standard H&E-stained images. This method is based primarily on the interaction or intercellular distance between tumor cells, which is a surrogate for intercellular adhesion. It is grounded in cancer biology, wherein neoplastic cells within a tumor that lose cell-to-cell adhesive properties are more capable of invasion through the extracellular matrix and finally into lymphovascular spaces to form metastases.

Traditional nuclear grading is based on visual inspection of nucleolar size. The overall assigned grade is based on the highest-grade focus that may span only a few high-power microscopic fields. Consequently, there is considerable grade heterogeneity, and ISUP/WHO grading can under-grade smaller samples such as biopsy material. Our findings of intercellular interaction distance were based on the analysis of three small regions of interest per patient, spanning approximately 500 µm × 500 µm each. This would equate to three microscopic high-power 0.5 mm × 0.5 mm fields, an amount of tissue that most renal mass biopsies would readily accommodate.

The observed inhibition or spatial randomness pattern in Group 1, especially after 8 µm, indicates a deviation from indolent primary tumor behavior, in which cell adhesion is critical in maintaining the tumor mass. The metastatic cascade primarily involves the breakdown of intercellular adhesion from the underlying lamina propria, enabling malignant cells to depart from their original location and adopt a more mobile and invasive phenotype, ultimately initiating invasion and metastasis [22]. The significantly higher interaction distances found in Group 1 patients, in conjunction with PCF analysis, support this logic. Moreover, modeling the intercellular interactions between tumor cells in patients in Group 1 and those with metastatic disease compared to interactions between random cells further reinforces this conclusion.

In the context of metastasis, the complete set of extracellular matrix molecules, the matrisome, becomes particularly relevant. Specific components of the matrisome such as collagen, fibronectin, and proteoglycans can facilitate the invasion of cancer cells through surrounding tissues [23,24]. Given the matrisome’s critical role in promoting metastasis, it is notable that our pathway analysis on highly differentially expressed genes yielded a core matrisome signature comprising 21 genes. Examining this gene list sheds light on potential molecular mechanisms underlying the altered cell adhesion and metastatic progression. Several genes from the provided list stand out for their known roles in cell adhesion and the metastatic cascade.

The gene VWC2 (von Willebrand factor C domain-containing protein 2) is implicated in cell adhesion processes [25]. Its expression increases tumor-platelet heteroaggregates and confers enhanced metastatic activity [26], and alterations in VWC2 expression may impact the adhesive properties of cells, potentially influencing metastatic behavior. PRSS3 (serine protease 3) is known for its involvement in extracellular matrix remodeling, a crucial step in metastatic progression [27]. In pancreatic cancer, secreted PRSS3 activity is linked to increased invasiveness of cancer cells by facilitating the degradation of the extracellular matrix [28], enabling cells to invade surrounding tissues. WNT7A, a member of the Wnt signaling pathway, plays a pivotal role in cell adhesion and invasion [29]. WNT7A can upregulate expression of the cell adhesion-related protein fibronectin [30], which could contribute to the inhibition of cell adhesion in metastatic disease. In melanoma and lung cancer cell lines, increased expression of MMP8 (matrix metalloproteinase 8) prevents metastasis formation through the modulation of tumor cell adhesion and invasion [31]. Downregulation of MMP8 expression (−4.0), as seen in our study, may contribute to the disruption of intercellular adhesion, facilitating the migration of cancer cells and promoting metastatic spread.

These genes and others in the list of core matrisome genes, such as MUC15, MATN4, and FGL1, have known functions related to cell adhesion, extracellular matrix modulation, epithelial–mesenchymal transition, and metastatic processes [32,33,34]. Further investigation into the specific activity of these genes will provide molecular insights into the observed spatial distribution patterns and enhanced cell-to-cell interactions, offering a more comprehensive understanding of the underlying mechanisms driving metastasis.

The aggressive nature of ccRCC disease in a larger cohort of 352 patients with altered gene expression among our subset of highly differentially expressed genes augments our narrative on the potential functional consequences of these genes. In line with our reasoning, alteration in these genes could drive aggressive behavior and metastasis, as reflected in the higher disease stage and poorer progression-free survival in gene-altered patients.

Limitations of our study include the relatively small number of cases as well as the fact that we used images from RCC resections as input material. We chose the initial set of *n* = 72 samples since they were derived from a single institution. The workflow to generate the H&E-stained slide images—from retrieval from the OR, tissue fixation, processing and staining—was uniform and thus other confounding factors related to pre-analytic variables were minimized. We did, however, validate our gene expression signature derived from the spatial analysis on a larger multi-institutional cohort of 352 ccRCC patients. Our signature was able to stratify the foregoing patients by stage and survival, thereby attesting to its robustness in a larger, multi-institutional cohort. Nonetheless, validation on larger ccRCC cohorts could enhance the generalizability of our findings. Additionally, the gene expression differences between spatial groups are based on bulk sequencing of tumor and the tumor microenvironment and do not reflect the underlying spatial heterogeneity of the cells. Future studies should consider incorporating advancements in spatial transcriptomics technologies to investigate the spatial expression heterogeneity of these genes. Finally, the matrisomal signature was used in this context to confirm the biological underpinnings of our spatial analysis. However, this specific signature will need to be functionally validated in ccRCC model systems in subsequent biomarker driven follow-up studies.

Analyzing renal mass biopsy material is indicated because clinical decision making and prognostication are ideally carried out before tumor resection. Moreover, measurement of interaction distance requires cells to be segmented, which means their nuclei must be traced either manually or with some pre-trained algorithms. In this study, this process was carried out manually by annotating and training a model that was unique to each specific histology color and pattern. This is labor intensive and serves more as proof of principle than as a technique that can be applied to clinical practice. With greater AI segmentation capability, this process can be streamlined in the future for integration into clinical pathology practice.

## 5. Conclusions

In conclusion, this study contributes valuable insights into the association between the spatial distribution of tumor cells and metastasis in ccRCC. The identified spatial association and related gene signature of ccRCC form the basis for future investigations into novel prognostication models derived from H&E-stained images in the hope of advancing personalized medicine approaches for patients.

## Figures and Tables

**Figure 1 cancers-17-00249-f001:**
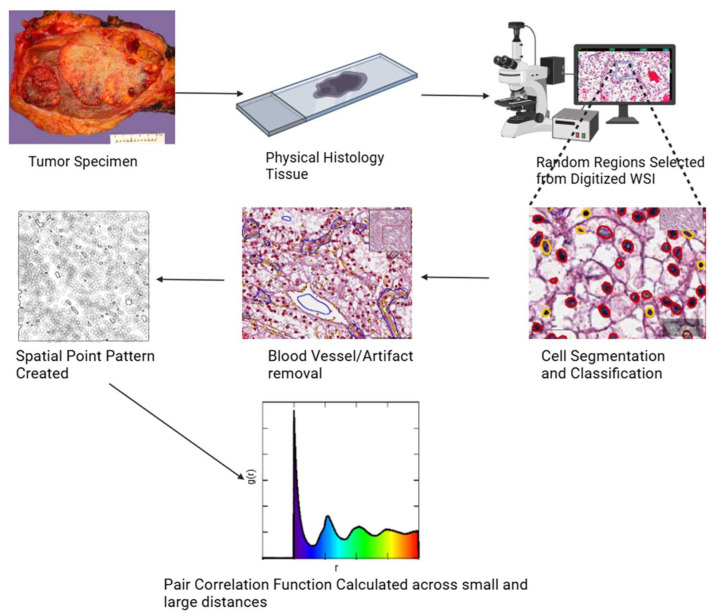
Study workflow from tissue selection to modeling data and gathering pair-correlation function statistics.

**Figure 2 cancers-17-00249-f002:**
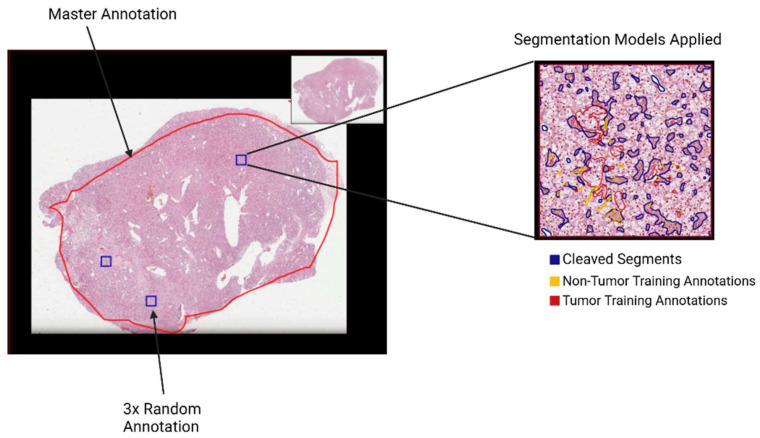
Pathologist’s annotation of tissues. Different levels of annotations were used to identify, classify, and segment the tumor (TCGA-A3-3343 used as an example).

**Figure 3 cancers-17-00249-f003:**
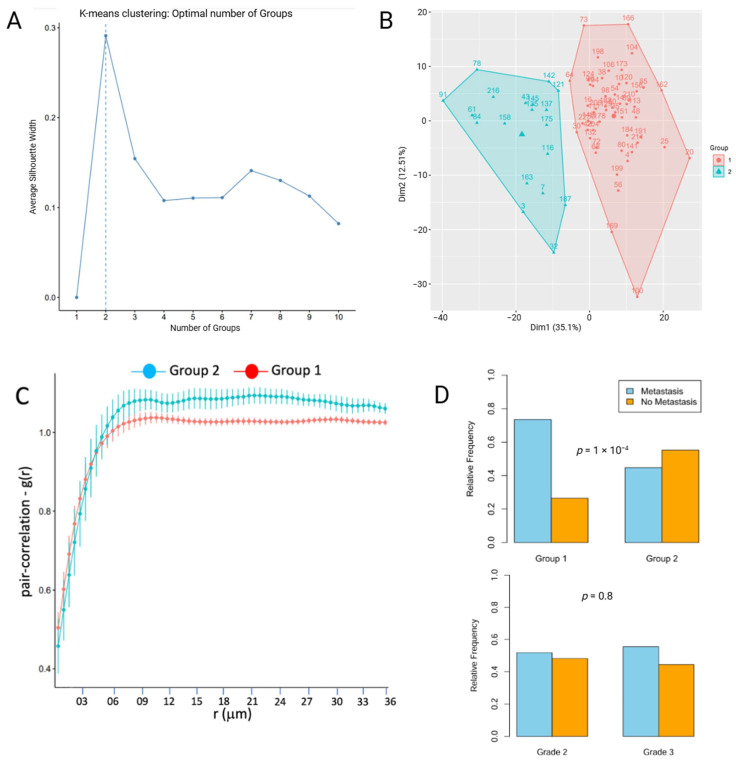
(**A**) Optimal number of K-means clusters (dashed line). (**B**) Principal component analysis of the PCF between the two patient groups. (**C**) PCF between the 2 groups. (**D**) Association of metastasis with respect to spatially defined groups 1 and 2 versus ISUP/WHO grade 2 and 3.

**Figure 4 cancers-17-00249-f004:**
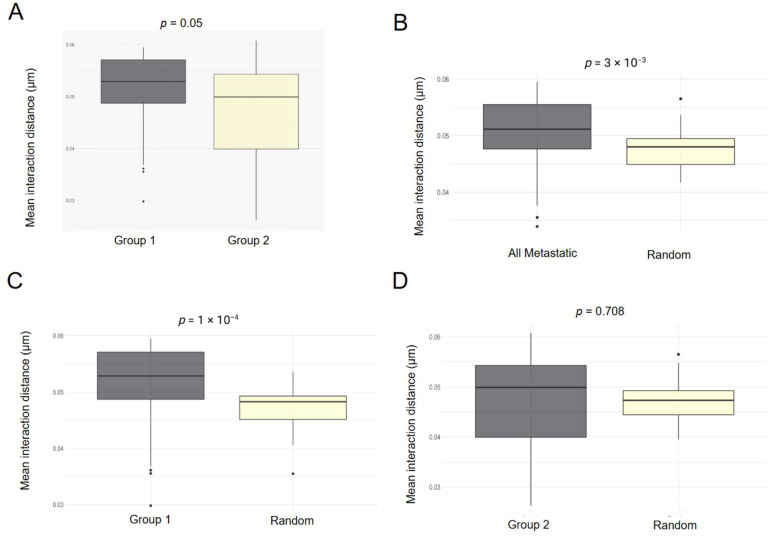
Parameters of Geyer models on spatial clusters. (**A**) Mean interaction distances in Group 1 and Group 2. (**B**) Mean interaction in between all metastatic cases compared with simulated patterns of spatial randomness. (**C**,**D**) Mean interaction distances in Group 1 and Group 2, respectively, compared with simulated patterns of spatial randomness.

**Figure 5 cancers-17-00249-f005:**
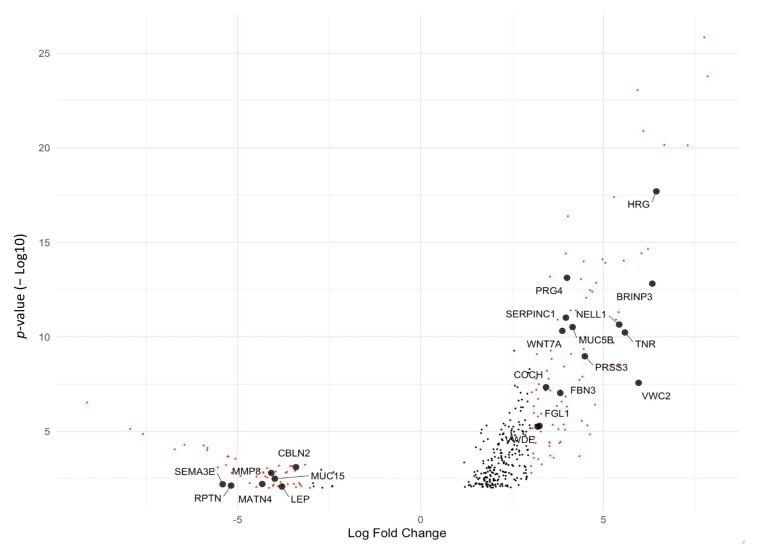
Fold change versus *p*-value for differentially expressed genes between Group 1 and Group 2 patients (adjusted *p*-value cutoff false discovery rate < 0.01). Also highlighted are the matrisomal genes with fold change greater than 3.

**Figure 6 cancers-17-00249-f006:**
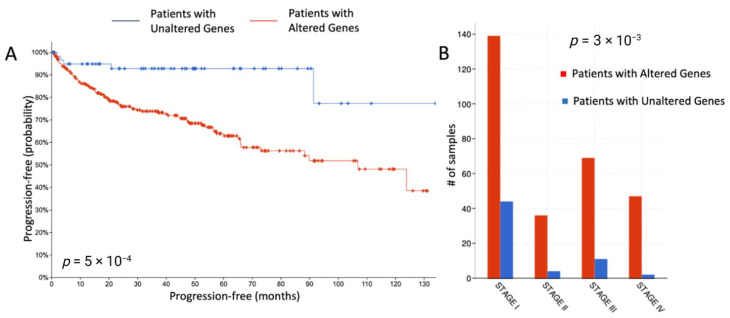
(**A**) Progression-free survival in a multi-institutional TCGA cohort of ccRCC patients with altered and unaltered genes. (**B**) Disease stage segregation of samples with altered and unaltered genes.

**Table 1 cancers-17-00249-t001:** Clinical and pathologic characteristics of patients with non-metastatic and metastatic clear cell renal cell carcinoma (ccRCC).

Clinicopathologic Features	Non-Metastatic ccRCC	Metastatic ccRCC
ISUP/WHO Grade 2	10	8
ISUP/WHO Grade 3	28	26
Pathologic Stage T1–T2	20	10
Pathologic Stage T3–T4	18	24
OS status ^1^	61%	26%

^1^ OS, overall survival.

## Data Availability

All the data and the codes are available at https://github.com/PrahaBhat/RCC-project.git [accessed on 8 January 2025]. The original slide images can be obtained from the TCGA GDC portal matching the TCGA IDs provided in Appendix A. We obtained H&E-stained images from 72 patients, nearly half of whom had metastatic disease. Table 1 summarizes the patients’ tumor grade, pathologic stage, survival status, and metastatic status. Appendix A lists the specific TCGA identifiers. Appendix A provides granular clinical data, which is also available via the TCGA GDC portal.

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
