# Peer review of "Spatial Distribution of Tumor Cells in Clear Cell Renal Cell Carcinoma Is Associated with Metastasis and a Matrisome Gene Expression Signature"

_cancers, 2025, doi:10.3390/cancers17020249_

Round 1
Reviewer 1 Report
Comments and Suggestions for Authors
The paper “Spatial Distribution of Tumor Cells in Clear Cell Renal Cell Carcinoma is Associated with Metastasis and a Matrisome Gene Expression Signature” by Bhat et al addresses an important question of assessing RCC aggressiveness based on H&E staining. The study is well written and easy to comprehend. The authors tested the potential of spatial point process modelling analysis for prediction of metastasis in clear cell renal cell carcinoma. The authors demonstrate that spatial analysis of hematoxylin – eosin stained images can identify two distinct groups with different risks of metastasis development. Further the authors show that this way of risk assessment is more efficient that the classical tumor grading. The authors identified matrisome gene expression signature that is associated with the identified spatial patterns. metastatic risks, and that this approach functions better than classical tumor grading. Additionally, the author identified a matrisome gene expression signature associated with this spatial patterns.
Introduction provides sufficient information on the topic studied in the paper. Methods are described clearly. Results section is easy to read and the figures are clear and informative. Discussion compares obtained results with state of the art research and present the limitations of the study.
Overall, this is an interesting study that demonstrates the potential of spatial point analysis for prognosis determination of RCC, though further research is needed to validate this approach.
Specific comments
1. More detailed clinical information should be provided if possible, including treatment received by the subjects and sites of metastasis.
2. Taking into account the relative simplicity of the experimental part of the research (H&E staining), the number of clinical samples of 72 is rather small. The authors do recognize it as a limitation of the study, however it is unclear why the larger sample or validation of the finding on another sample was not possible.
3. The results of expression profiling should be confirmed by different method, like qPCR on the same or another cohort of patients.
4. As well though the authors identify a matrisome gene expression signature, further studies are needed to validate the functional role of these genes in metastasis and spatial pattern formation. This should be at least discussed in detail.
Author Response
Reviewer # 1:
Specific comments
- More detailed clinical information should be provided if possible, including treatment received by the subjects and sites of metastasis.
A1. We thank the reviewer for his helpful suggestions. We have provided granular clinical data from TCGA in a new Sup Table 6 in the Data Availability Statement. TCGA samples did not receive any neoadjuvant therapy as it was an exclusion criterion; however, any adjuvant therapy is indicated for these patients (column GO = treatment_type). Detailed clinical and pathologic data are also provided in Sup Table 6, including the presence of metastasis (Column AD = ajcc_pathologic_m); the specific metastatic sites, however, are not provided by TCGA (Column CY (metastasis_at_diagnosis_site)).
- Taking into account the relative simplicity of the experimental part of the research (H&E staining), the number of clinical samples of 72 is rather small. The authors do recognize it as a limitation of the study, however it is unclear why the larger sample or validation of the finding on another sample was not possible.
A2. We agree with the reviewer that this point requires further explanation. We have added text in page 13 (Discussion) as follows: “We chose the initial set of N=72 samples since they were derived from a single institution. The workflow to generate the H&E stained slide images—from retrieval from the OR; tissue fixation, processing and staining--was uniform and thus other confounding factors related to pre-analytic variables was minimized. We did, however, validate our gene expression signature derived from the spatial analysis on a larger multi-institutional cohort of 352 ccRCC patients. Our signature was able to stratify the foregoing patients by stage and survival, thereby attesting to its robustness in a larger, multi-institutional cohort.”
- The results of expression profiling should be confirmed by different method, like qPCR on the same or another cohort of patients.
A3. We agree with the reviewer that validation of the expression signature is important for biomarker driven studies. However, the scope of this article is computational and the importance of using spatial analytics in delineating differences in the behavior of ccRCC derived from analysis of standard H&E images. The matrisomal signature was used to confirm the biological underpinnings of our spatial analysis rather than functionally validating specific gene expression signatures.
- As well though the authors identify a matrisome gene expression signature, further studies are needed to validate the functional role of these genes in metastasis and spatial pattern formation. This should be at least discussed in detail.
A4. We agree with the reviewer that our results will require functional validation in in vitro and in vivo model systems of ccRCC for biomarker driven studies. This would be appropriate in follow up studies that address the biology of the matrisomal expression signature.
Per request of the reviewer, we have added text to elaborate on this point in our Discussion, page 13, as follows:
“Finally, the matrisomal signature was used in this context to confirm the biological underpinnings of our spatial analysis. However, this specific signature will need to be functionally validated in ccRCC model systems in subsequent biomarker driven follow-up studies.”
Reviewer 2 Report
Comments and Suggestions for Authors
The modern grading system is not perfect in predicting the malignant characteristics of clear cell renal cell carcinoma (ccRCC). The authors propose an elegant method for characterizing ccRCC by analyzing the spatial distribution of tumor cells (i.e., the distance between individual cells) in H&E-stained tissue sections. They retrospectively examined H&E-stained images from 72 grade 2 and 3 ccRCC patients and demonstrated that greater spatial randomness was more strongly associated with metastasis. Furthermore, they performed differential gene expression analysis, revealing a gene expression signature related to the extracellular matrix that correlated with both the spatial patterns and the aggressive behavior of ccRCC. The validation set included data from 352 ccRCC patients across multiple institutions, sourced from the cBioPortal.
The manuscript is well-written and includes six figures, citing 34 references, which is adequate for a study of this scope. It also contains one table. The data presented in this manuscript could have a direct impact on the pathological diagnosis of ccRCC in the future. Additionally, the manuscript provides valuable insights for basic research on epithelial-mesenchymal transition in RCC.
There is one minor issue to address: the authors should include the column titles in Table 1. Additionally, it would be helpful to include percentages for overall survival (OS) status in Table 1 (e.g., % living, % dead) to improve reader comprehension.
Author Response
There is one minor issue to address: the authors should include the column titles in Table 1. Additionally, it would be helpful to include percentages for overall survival (OS) status in Table 1 (e.g., % living, % dead) to improve reader comprehension.
A1. We thank the reviewer for his helpful comments, and we have modified Table 1 accordingly.